# 1    Monitoring European anthropogenic NO$_x$ emissions from space

Ronald J. van der A[1*], Jieying Ding[1*], Henk Eskes[1]
[1]Royal Netherlands Meteorological Institute (KNMI), De Bilt, The Netherlands
[*]Corresponding authors: Ronald van der A (avander@knmi.nl), Jieying Ding (jieying.ding@knmi.nl)
**Abstract**
Since the launch of TROPOMI on the S5p satellite, NO$_2$ observations have become available
with a resolution of 3.5x5 km, which makes monitoring NO$_x$ emissions possible at the scale of
city districts and industrial facilities. For Europe, emissions are reported on an annual basis for
country totals and large industrial facilities and made publicly available via the European
Environmental Agency (EEA). Satellite observations can provide independent and more timely
information on NO$_x$ emissions. A new version of the inversion algorithm DECSO (Daily
Emissions Constraint by Satellite Observations) has been developed for deriving emissions for
Europe on a daily basis, averaged to monthly mean maps. The estimated precision of these
monthly emissions is about 25% for individual grid cells. These satellite-derived emissions
from DECSO have been compared to the officially reported European emissions and spatial-
temporal disaggregated emission inventories. The country total DECSO NO$_x$ emissions are
close to the reported emissions and the emissions compiled by the Copernicus Atmospheric
Monitoring Service (CAMS). The comparison of the spatial distributed NO$_x$ emissions of DECSO
and CAMS showed that the satellite-derived emissions are often higher in cities, while similar
for large power plants and slightly lower in rural areas.

## 1. Introduction

Nitrogen oxides (NO$_x$) concentrations play an important role in air quality, the nitrogen cycle,
and as precursor for climate gasses, knowledge of NO$_x$ emissions is also important for climate
studies (Shindell et al., 2005). Because of the importance of NO$_x$ for air quality, in Europe both
the concentrations in air and emissions to air are regulated. Country total NO$_x$ emissions need
to be reported by EU countries as part of the Convention for Long-Range Transboundary Air
Pollution (LRTAP, Pinterits et al., 2021) and the National Emission reductions Commitments
(NEC) Directive (NEC, 2023) of the European Union. More detailed emission inventories
including spatial distribution are compiled based on reported emissions, statistical
information (e.g. population density) and activity data. Examples of these inventories on a
global scale are the Emissions Database for Global Atmospheric Research (EDGAR, EC-JRC/PBL,
2011, Janssens-Maenhout et al., 2015) and the various global and regional emission
inventories developed in the context of the Copernicus Atmosphere Monitoring Service
(CAMS, Innes et al, 2019) of the EU Copernicus programme. These gridded emission
inventories are widely used for global atmospheric composition and regional air quality
modelling. The realism of the air quality model results depends largely on the accuracy of the
emission inventory (Thunis et al, 2021).
Since the availability of satellites capable of measuring $NO_2$ concentrations in the atmosphere,
methods have been developed to derive top-down emissions (Streets et al., 2013). These top-
down emissions have the major advantage that they are based on observations. This fully
independent source of information provides the possibility to check reported emissions,
monitor rapid changes (e.g. due to the COVID-19 lockdowns) and has the potential of finding
unknown and unreported sources. Polar-orbiting satellites with a global daily coverage within
1-3 days, allow monitoring of changes in emissions on timescales of days to weeks. Nadir-
viewing satellites measure total column concentrations of trace gases, and the distinction of
source sector type must be deduced via the source location. A popular inversion technique
for $NO_x$ emissions is the divergence method of Beirle et al. (2021, 2023), where the average
flux is calculated in grid cells, assuming local mass balance, to find the sources of the
emissions. Although no model is needed in this method, the required spatial derivations lead
to noisy fields for daily overpasses, and it only provides useful emissions when averaged over
a longer period. Furthermore, assumptions must be made for the chemical lifetime, and
simplifications lead to biases, especially in background emissions. A second class of methods
is based on plume fitting (Fioletov et al., 2022). This method can be applied to individual
overpasses but needs well-defined plume shapes which is not trivial for areas with multiple
sources close together. Both these methods simplify atmospheric transport as two-
dimensional. For a full three-dimensional description of transport and chemistry, a data
assimilation or inverse modelling method is used to match the model results and observations
by adapting the emissions (Miyazaki et al., 2017, Fortems-Cheiney et al., 2021). A typical
application of satellite-derived emissions is the study of the impact of recent events, for
example the effect of COVID regulations (Ding et al., 2020). Top-down emissions are also used

for the verification and support to improve current emission inventories (Guevara et al., 2021; Crippa et al., 2023). Guevara et al. (2021) and Cripa et al. (2023) concluded that interesting aspects for future studies are the spatial distribution, seasonal time profiles and multi-annual trends of the emissions.

In this study we present the latest version 6.3 of the Daily Emissions Constrained by Satellite Observations DECSO (DECSO) inversion algorithm. The DECSO algorithm can be applied for the operational monthly (or even daily) monitoring of emissions for any region worldwide based on satellite observations of trace gases such as $SO_2$, $NH_3$ or $NO_2$. In this paper this new DECSO version has been applied to $NO_2$ observations over Europe from the TROPOMI instrument (Veefkind et al., 2012) on board the Sentinel-5P satellite. The DECSO system is efficient, requires only a single forward run of the chemistry-transport model and takes about 12 hours to process one month of data on a 30-core computer. Here, we will evaluate the performance of DECSO on various spatial scales (from national to point sources) by comparison with the various bottom-up emission inventories available for Europe. By comparing satellite derived emissions with bottom-up emissions we gain insight in the accuracy of both derived emission datasets.

## 2. Methodology and data

### 2.1 DECSO: inversion of TROPOMI observations

The inversion algorithm DECSO (Daily Emissions Constrained by Satellite Observations) has been developed at KNMI for the purpose of deriving emissions for short-lived gases (Mijling and van der A, 2012). DECSO is using a Kalman Filter implementation for assimilating emissions. The emission forecast model is based on persistency from the analysis, while the concentrations are calculated from the emissions by a chemical transfer model (CTM) and compared to satellite observations. The sensitivity of concentrations to emissions is calculated from multiple forward trajectories to account for the transport of the short-lived gas, but only a single CTM forward run is needed. More detailed information on the method can be found in Mijling and Van der A (2012), the validation is described in Ding et al. (2017a) and the previous latest published version, i.e. DECSO v5.2, is described in Ding et al. (2020). Recent developments of the algorithm to improve its resolution and quality have led to the release

of version 6.3. The most important updates are the use of a recent version of the chemical
transport model, improved use of TROPOMI observations and changes in the sensitivity matrix
calculations. More details of these updates follow below.
The chemical transport model in DECSO has been upgraded to the latest version of the
Eulerian regional off-line CTM CHIMERE v2020r3 (Menut et al., 2021). The implementation of
CHIMERE in DECSO was described in Ding et al. (2017b). In this study CHIMERE is combined
with the Copernicus Landcover 2019 data (Buchhorn et al., 2020) and HTAP v3 (Hemispheric
Transport of Air Pollution, Crippa et al., 2023) of 2018 for the source sector split of the
emissions. The meteorological input data for CHIMERE are the operational European Centre
for Medium-Range Weather Forecasts (ECMWF) weather forecasts.
The sensitivity matrix, giving the relationship between emissions and concentrations, is based
on trajectories calculated with a high temporal resolution (a time step of 7.5 minutes). In the
new version the relationship between emissions and concentrations is limited to a maximum
distance of 150 km to avoid effects of errors in the trajectories over longer distances. With
this sensitivity matrix not only observations over the source are affecting the derived
emissions, but also the transported concentrations away from the source within 150 km. The
default settings of DECSO described here are for a grid resolution of 0.2 degree. For higher
grid resolutions, the settings for temporal resolution and maximum trajectory distance are
increased and reduced respectively.
The error parametrizations for the emission model and observations are based on the
Observation-minus-Forecast (OmF) and the Observation-minus-Analysis (OmA) statistics of
previous runs. The latest version of DECSO can also be applied to simultaneous optimisation
of emissions of $NO_x$ and $NH_3$ (Ding et al., 2024).
Although HTAP v3 has been used for the sector distribution of emissions and other species in
CHIMERE, no use is made of a -priori (bottom-up) $NO_x$ emissions in DECSO. DECSO is using a
persistency forward model in which the emissions of the current day are equal to the
emissions of the previous day. In addition, there is a strong dependency of the calculated
emissions on the observations as shown in Ding et al. (2021). Since the derived emissions are
updated by addition and not by multiplication factors, unknown sources or emission changes
are detected fast.
TROPOMI is a spectrometer instrument onboard the Sentinel 5P satellite, which was launched
in October 2017 and is flying a sun-synchronous polar orbit with a local overpass time of 13:30.
The measured NO$_2$ columns are derived from the visible band that has a spectral resolution
of 0.54 nm (0.2nm sampling) and a signal-to-noise ratio of about 1500 (van Geffen et al.,
2022a). The NO$_2$ tropospheric columns have a spatial resolution of 5.5 x 7 km (5.5 x 3.5 km
since 6 August 2019) over a swath of about 2600 km, which means that global coverage is
reached daily.
We are using the latest version 2.4 reprocessed and offline TROPOMI NO$_2$ observations (van
Geffen et al,2022b) converted to super-observations as described in Ding et al. (2020). The
modelling of NO$_2$ in the free troposphere, governed by processes like lightning, deep
convection, aircraft emissions or long-range transport, is often simplified in regional air-quality
models focusing on surface concentrations. However, the TROPOMI NO$_2$ product is providing
a tropospheric column, which includes the Planetary Boundary Layer (PBL) and the free
troposphere. As a result, model biases in the free troposphere may be a significant source of
systematic error in the model-satellite comparisons (Douros et al., 2023). To mitigate this
problem we adapt the TROPOMI NO$_2$ retrieval by calculating a partial column up to the 700
hPa level instead of the tropopause level. The stratosphere + free troposphere NO$_2$ column
from the TM5-MP (Tracer Model 5, https://tm5.site.pro/, Williams et al., 2017) assimilation
system are now subtracted from the satellite-observed total column, and new retrieved layer
column amounts, air-mass factors and kernels are computed for the surface to 700 hPa layer
in the same way as they are computed for the tropospheric column (van Geffen et al., 2022b).
The observations with a cloud radiance fraction of more than 50% (this corresponds to a cloud
fraction of about 20%) have not been used. For Europe, it means that about 45% of the
observations are used.
Superobservations (Sekiya et al., 2022) are constructed as the area-weighted mean of cloud-
free (qa value > 0.75) TROPOMI observations over the CHIMERE model grid cells. For a grid of
0.2x0.2 degree a superobservation contains about 10 to 15 TROPOMI NO$_2$ observations. The
use of superobservations improves the signal-to-noise ratio and it reduces the calculation time
of DECSO. On the other hand, the sampling of transported NO$_2$ from the observations
calculated back to the source on the emission grid, based on superobservations, will slightly
spread out the derived emissions and reduce their spatial resolution compared to using
individual observations. The chosen size of the superobservation grid of 0.2x0.2 degree is
therefore a compromise between noise, calculation speed and spatial resolution. Knowing
that the smoothing of emissions after averaging can be imagined as a distribution by a pyramid

shape weighting function around a point source, a deconvolution is possible for isolated emission sources with a known location. The current version of DECSO makes use of the superobservations software as also used in Sekiya et al. 2022. The software has been further developed focusing on a realistic description of the superobservation uncertainty (Rijsdijk et al, 2024) and this new superobservation software is planned to be used in future DECSO studies.

In a post-processing step, the total monthly $NO_x$ emissions are split into anthropogenic and (biogenic) soil emission contributions Lin et al. (2023). The soil emissions show a strong seasonal cycle with low emissions in winter, while the anthropogenic emissions are more constant over the year. The soil $NO_x$ emissions are derived by fitting the monthly emissions in a selection of grid-cells without any significant anthropogenic contribution according to land-use data. In this way the monthly averaged soil $NO_x$ emissions in the categories for forest, agricultural and shrub-land are derived. These monthly soil $NO_x$ emissions are weighted with the land-use type of these 3 categories in each grid cell and subtracted from the total derived $NO_x$ emissions to end up with the anthropogenic $NO_x$ emissions discussed in this study. This splitting method is described in detail in Lin et al. (2023).

For the monthly emissions also the precision of the emission in each grid cell has been calculated. Each daily $NO_x$ emission per grid cell derived by DECSO is accompanied by a standard deviation calculated according the Kalman Filter equations (the standard deviation is part of the emission data product of DECSO). As the starting point of each daily step in the calculation by DECSO is the emissions of the previous day, the resulting emissions will show an autocorrelation in their errors. For each grid cell the autocorrelation function $\rho_k$ (for time lag $k$) has been calculated for each month. We see typically that the autocorrelation effects in the errors have disappeared completely after about 1 week.

When calculating the variance of the monthly mean values, we must take this autocorrelation function into account. The variance $S$ of the monthly mean $NO_x$ emissions per grid cell is calculated following Bayley and Hammersley (1946) or Box et al. (2008) as

$$S = \frac{\sigma^2}{n}\left[1 + 2\sum_{k=1}^{n-1}\left(1 - \frac{k}{n}\right)\rho_k\right] \, ,$$

where σ is the mean standard deviation of the emissions over the month and *n* is the number
of days in the month. We assume here that σ is not varying a lot over the month. This precision
σ is calculated in the Kalman equations of the inverse modelling and it depends on the
precision of the TROPOMI $NO_2$ superobservations. The precision depends on the location and
emission magnitude, but on average the precision is estimated as 8% for annual emissions,
25% for monthly emissions and between 10 and 60 % for the daily emissions.

In this study we will focus only on $NO_x$ emissions. Although DECSO has been applied to many
regions in the world, we will show results for a domain over Europe (35°-55°N, 10°W-30°E) and
for 0.2 degree spatial resolution. The temporal resolution of our inversion is daily, usually
averaged to monthly or yearly mean values, for the period of 2019 to 2022. Figure 1 shows
the average annual emissions for 2019 as derived with DECSO version 6.3. In the Figure the
emissions of major cities and industrial facilities can be identified. Ship emissions show up
clearly in most seas where many ships follow the same route. Other areas over sea appear
noisier since ship locations are moving while emitting $NO_x$. The most polluted regions in
Europe are the densely populated and industrial regions in the Po Valley, the Ruhr area, and
the West of the Netherlands.

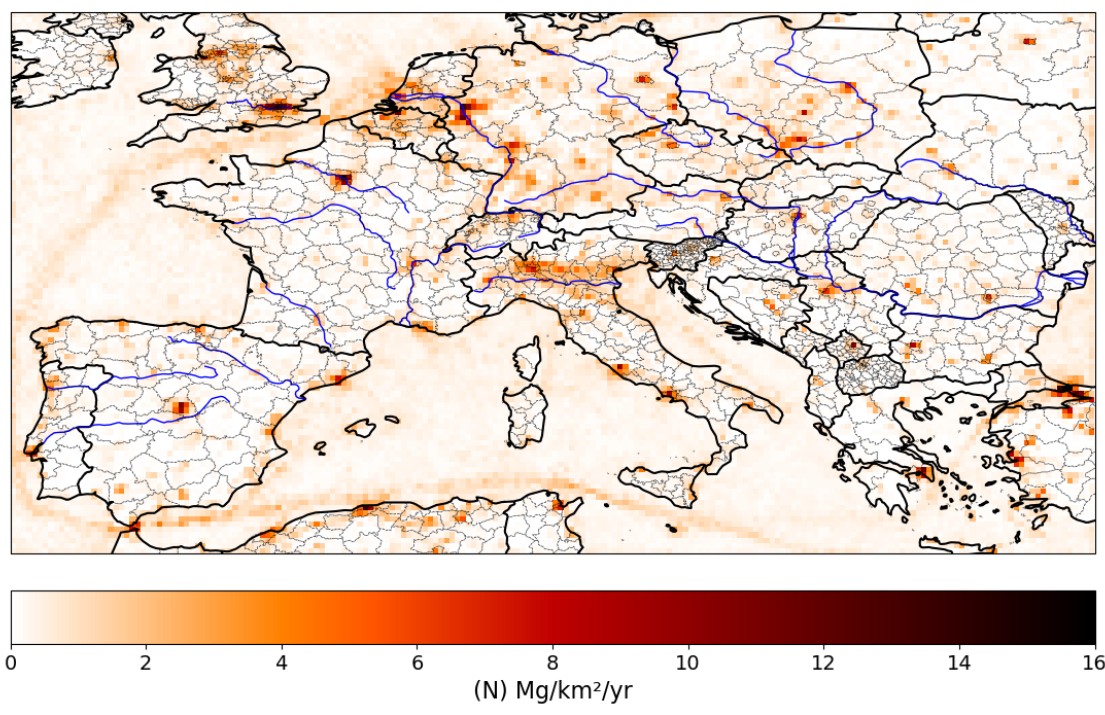

(N) Mg/km²/yr


**Figure 1**        The annual-averaged anthropogenic $NO_x$ emissions for 2019 derived from
211                            TROPOMI $NO_2$ observations using the DECSO algorithm.



**2.2 Databases for validation**
For comparison of the emission results in Europe we will use several inventories, all based on
official emissions reported to the European Environmental Agency (EEA). The first one is the
inventory of national emissions per source category reported under the National Emission
reductions Commitments (NEC) Directive of the European Union. Another similar inventory is
the Emission inventory reported under the Convention on Long-range Transboundary Air
Pollution (LRTAP), which give the country totals of emissions in various source categories. The
last one we will use is the European Pollutant Release and Transfer Register (E-PRTR; EPRTR,
2012), which is a database of the individual emissions of the biggest industrial facilities (above
0.1Mg/year) in Europe. The E-PRTR emissions data are reported on an annual basis. From here
on we will call those databases simply NEC, LRTAP and E-PRTR. Besides comparison with these
officially reported emissions, we will also compare our emissions to the regional
anthropogenic emission inventory CAMS-REG-ANT v5.1 for air quality in Europe (Kuenen et
al., 2022) developed for the Copernicus Atmospheric Monitoring Service (CAMS), hereafter
called CAMS-REG. For these annual CAMS-REG emissions we use the total emissions regridded
from $0.1° \times 0.05°$ to $0.2° \times 0.2°$ and exclude the soil emissions (i.e. agricultural categories), since
soil emissions are also excluded in DECSO. Temporal profiles are also derived in CAMS, which
allow us to compare timeseries for monthly averaged values. We will use the Copernicus
Atmosphere Monitoring Service TEMPOral profiles (CAMS-GLOB-TEMPO, Guevara et al.,
2021,2023) for comparison of monthly variations in anthropogenic $NO_x$ emissions. The global
emission data version 5.3, called CAMS-GLOB-TEMPO, on a resolution of $0.1° \times 0.1°$ has been
regridded to $0.2° \times 0.2°$ resolution and is hereafter referred to as CAMS-TEMPO.

**3. Evaluation of the satellite derived emissions**

**3.1 Country scale intercomparison**
The $NO_x$ emissions derived with DECSO have been summed over the countries in our domain
and compared to the registered total emissions in NEC and LRTAP. Note that for the national
total emissions the spatial resolution or spatial smoothing of the derived emissions play hardly
any role. In total 21 countries are completely covered by our geographical domain and have
reported their emissions. The total anthropogenic emissions (excluding soil emissions) for all
these 21 countries are 1.44 Tg/year according both LRTAP and NEC. The total calculated
anthropogenic emissions by DECSO are 1.54 Tg/year, about 7% higher than the reported
emissions. The total anthropogenic emissions of CAMS-REG (excluding soil emissions) for the
same region are 1.54 Tg/year, in agreement with DECSO. Note that the total soil emissions
derived by DECSO are 0.78 Tg/yr for the same region, but this number cannot be compared
because soil emissions in LRTAP and NEC are only given for the agricultural sector and not for
forestry. The anthropogenic country totals are shown in Figure 2. In general, we see a good
agreement with the official reported country total emissions of LRTAP and NEC except for Italy,
which has much lower reported emissions. Greece, on the other hand, has higher registered
emissions, but the mismatch might be related to the difficult counting over the Greek islands,
since we have weighted the emissions by the land fraction in each grid cells to exclude
maritime emissions in these country totals. For CAMS-REG we see bigger deviations not only
for Italy, but also for Germany, Poland, and Spain. Note that Ireland is only partly in our
geographical domain and has therefore lower emissions according to DECSO. Besides the
comparison on a national level also on a provincial scale good agreement is found, as has been
shown for Cataloniain the EC-project SEEDS.

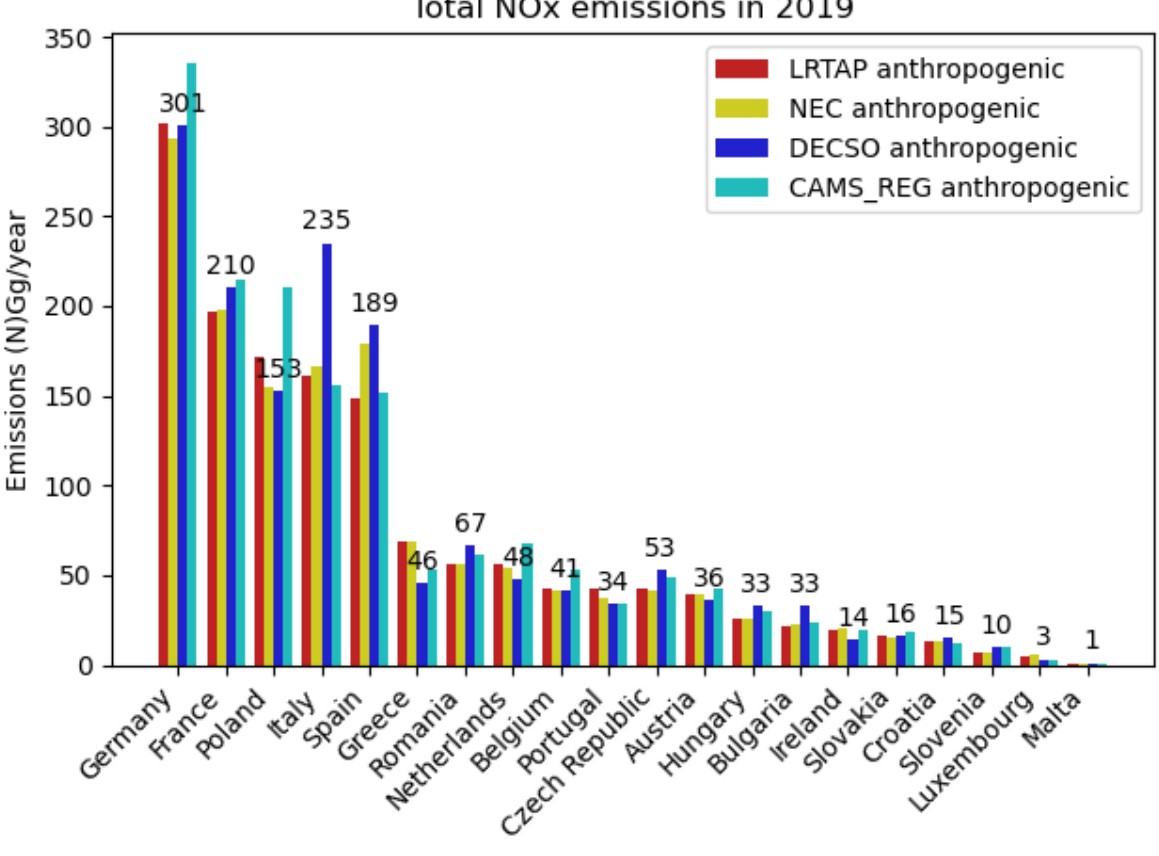


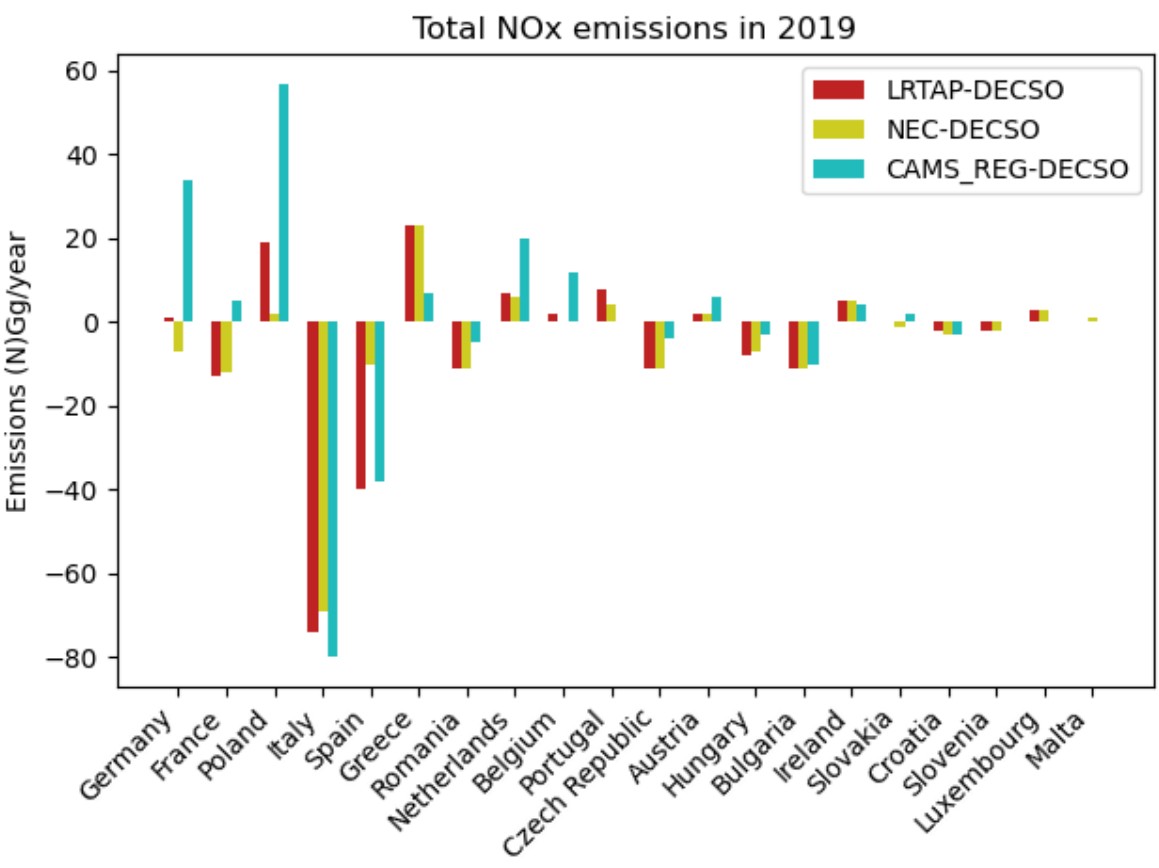


***Figure 2*** *(a) Country totals of anthropogenic NO$_x$ emissions (in (N)Gg/year) in the year*
*2019 according to databases LRTAP, NEC, CAMS-REG and the DECSO calculations. (b)*
*Differences in total emissions calculated by LRTAP, NEC, CAMS-REG compared to DECSO.*


**3.2 City scale**
With our current spatial resolution of 0.2x0.2 degree, we observe emissions per city district
for large cities, but the geographical distribution can be slightly blurred by the 0.2 degree
resolution of the TROPOMI superobservations. Figure 3 shows the spatial distribution of the
annual emissions of DECSO and CAMS-REG for three of the largest cities in Europe: Madrid,
Paris, and Rome. Although DECSO show similar emissions for the country totals, we see that
for large cities DECSO estimates higher emissions in the city center, and more activities are
seen in the region surrounding the city, as compared to the CAMS-REG emissions. The
industrial complexes at Rouen located north-west of Paris, and at the port of Civitavecchia
located west of Rome are similar in DECSO compared to CAMS-REG. The area of Rouen used
to have an active oil refinery, but in recent years the industrial emissions are about 0.11
(N)kg/km$^2$/h according to the E-PRTR database, which compares well to CAMS-REG and
DECSO. The spatial extent of high emissions in the Rome area is smaller in CAMS-REG, which
follows more the population density. However, the densely populated center of Rome is
surrounded by a busy ring road with a 20 km radius and a lot of commercial activities around
the city, which are not reflected in the population density map. The two powerplants at
Civitavecchia have reported emissions according to the E-PRTR database, which are equivalent
to about 0.17 (N)kg/km$^2$/h per grid cell, which is closer to the DECSO derived emissions.
Although this study focuses mainly on the land emissions, we see in the map for Rome, that
the maritime emissions of CAMS-REG and DECSO disagree a lot, and this is a topic for further
studies. The city emissions in Istanbul are much higher in DECSO than in CAMS-REG. These
emissions will include a lot of ship emissions since it includes the busy ship route through the
Bosporus Strait. The map of the greater area of London shows that DECSO has higher
emissions in the city, but lower outside the city. This is a pattern, we see in general: in most
big cities the emissions derived by DECSO show a similar distribution than in CAMS-REG but
the absolute emissions are higher, while the emissions in rural regions are usually lower in
DECSO than in CAMS-REG. The lower emissions in the rural regions can be seen in Figure S1,
which show maps for Europe of both emission products.
In Figure 4 we show the emission for two large industrial areas in Europe; the Po-Valley and
the Ruhr area. For the Po Valley the patterns are similar, but again the DECSO emissions are
higher in every city except for Genua in the Southeast corner of the map. For the Ruhr area,
the difference of emissions over the cities is small, the biggest differences are located at the
big power plants of Weisweiller, Neurath and Niederaussem around the open-pit lignite mine
of Hambach (the largest of Europe). The DECSO emissions are lower than CAMS-REG at the
locations of these power plants.

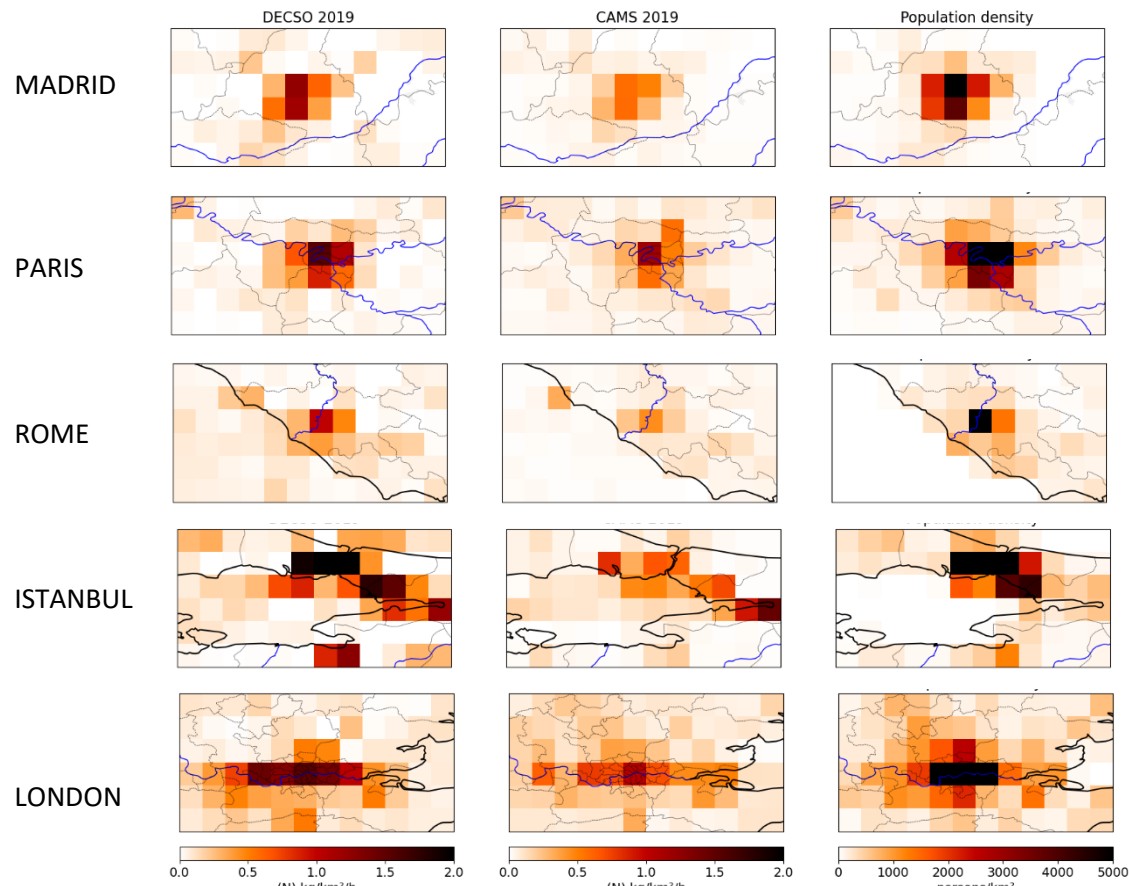

***Figure 3*** *Zoom-in plots for 5 large cities in Europe to illustrate the differences in*
*distribution of emissions of DECSO (first column), CAMS-REG (second column) and the*
*population density (third column) per km².*

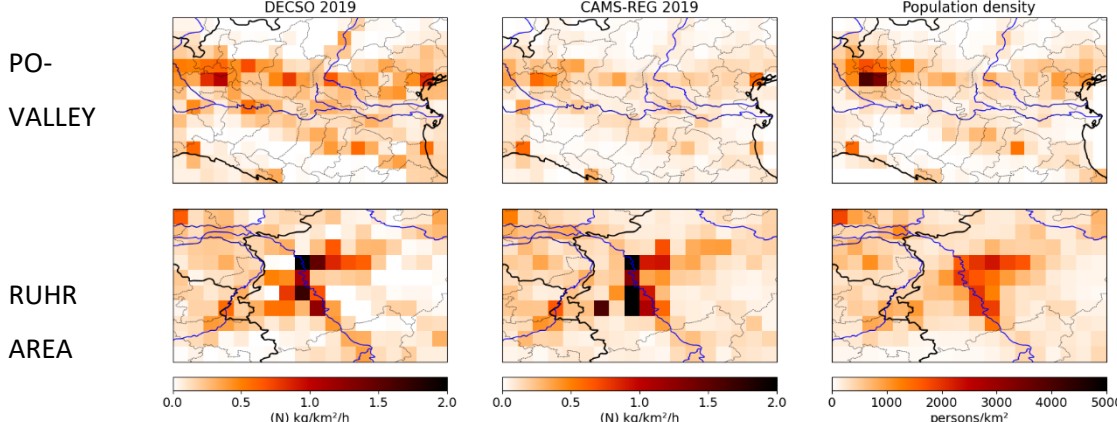


**Figure 4** *Zoom-in plots for two large densely populated and industrial regions in*
*Europe to illustrate the differences in distribution of emissions of DECSO (first column),*
*CAMS-REG (second column) and the population density (third column) per km².*


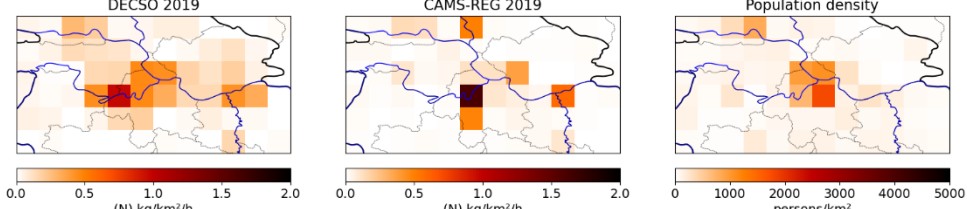

**Figure 5** *A map of North Serbia with NO$_x$ emissions of DECSO, and CAMS-REG. The*
*population map shows especially the higher population for Belgrade. The emissions in DECSO*
*are mainly correlated with the locations of several coal power plants (Nikola Tesla -A, -B, and*
*-Kolubara) and a cement factory (Lafarge in Beocin) in the North-West.*


On a European scale the biggest difference between CAMS-REG and DECSO was found for the
region around Belgrade in Serbia (Figure 5). The city of Belgrade is identified by the higher
population density in Figure 5. West of the city, the Nicola Tesla power plants are located,
which are strong emitters according to the E-PRTR database. They show up as a strong
emission source in the DECSO emissions, but they are mislocated in the current CAMS-REG
emissions.

327

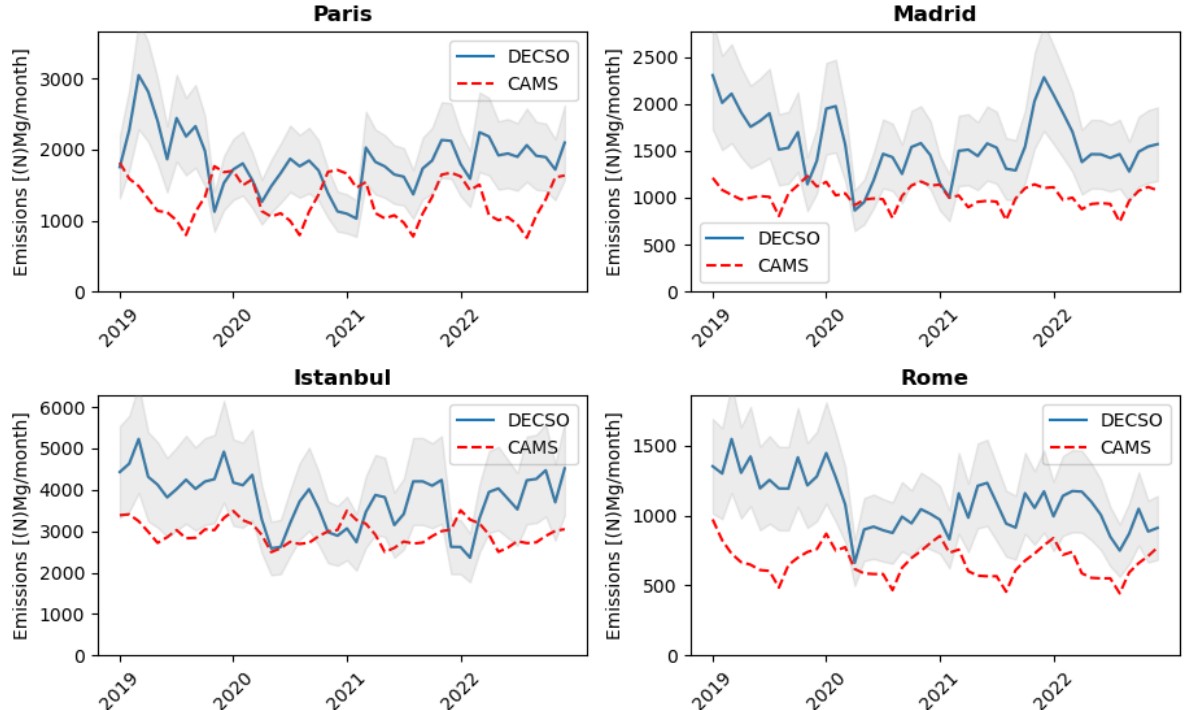

328

**Figure 6** *Timeseries of monthly NO$_x$ emissions derived by DECSO for the cities Paris, Madrid, Istanbul and Rome in the period 2019 to 2022. The shaded grey area shows the estimated uncertainty on the DECSO emissions. The dotted red line shows the CAMS-TEMPO NO$_x$ emissions for the same grid boxes.*

Figure 6 shows examples of timeseries for city emissions, in this case for the cities of Paris, Madrid, Istanbul and Rome (also shown in Figure 3). In these plots we report the total emissions in a square area of 5 by 5 grid cells centred on the city centre to make sure the whole city has been captured. As we had seen earlier, the DECSO emissions are on average higher than for CAMS-TEMPO, but also the seasonal cycle is different. The NO$_x$ emissions of CAMS-TEMPO show a seasonal cycle, which is almost identical each year, while DECSO show larger variations from year-to-year. We see clearly the effect of COVID regulations in all cities, that started first in March/April 2020 in Europe, and in the winter of 2020-2021 when strict COVID regulations were again in place. The general overall trend in this 4 year time period varies from city to city, but most cities show a slightly decreasing trend, partly related to a gradual decrease of emissions from road vehicles linked to European regulations.

**3.3 Intercomparison for large point sources**

To evaluate the performance of monitoring emissions from large point sources (LPS), we
compare the DECSO emissions with emissions registered in the E-PRTR data base. The isolated
LPS in Europe we selected are all large power plants close to lignite mines. Emissions from
DECSO are slightly spread to adjacent grid cells because the spatial resolution of the emission
field is less than the sampling of the grid cells as discussed in Sect. 2. To correct for this, we
can deconvolute the emissions around the isolated point source, but here we choose to sum
the anthropogenic emissions in the 3x3 grid cells including and around the point source to
make sure all emissions are accounted for. For the four cases discussed below, no significant
other sources exist in these 3x3 grid cell boxes, and soil emissions are excluded. The rural
anthropogenic emissions in such an area of 3x3 grid cells in Europe we estimate as about 0.13
(N)Gg/year by averaging the emissions of several similar rural 3x3 regions in Europe. We did
not correct for this background signal, but we included this in the error bars of Figure 7
The first case is that of the Maritsa Iztok facility in Bulgaria located next to an open coal mine.
There is no big city or any other industrial facility in the neighbourhood, except for the three
big power plants of the Maritsa Iztok facility. Figure 7 shows the monthly averaged emissions
calculated by the DECSO algorithm, the CAMS-TEMPO inventory, and the annual emissions
from the E-PRTR database for the Maritsa facility. For a fair comparison we selected for CAMS-
TEMPO also the same 3x3 grid cells around the LPS. For the period 2019-2022 the annual
emissions are given in Table 1 according to DECSO, CAMS-TEMPO and E-PRTR. The difference
in annual emissions between DECSO, CAMS-TEMPO and E-PRTR of the Maritsa facility are
within 20-40 %, although DECSO is the highest. The CAMS-TEMPO emissions show a negative
trend, which is not visible in DECSO that shows the highest emissions for 2022. Unfortunately,
no E-PRTR data for 2022 is yet publicly available.
The second power plant is the Bełchatów power plant in Poland with its capacity of 5,053 MW,
the biggest power plant of Europe. It is also one of the most polluting power plants in the
world and gets its fuel from the adjacent lignite coal mine of Bełchatów (Guevara et al., 2023).
For the year 2020 no emission values are reported in the current E-PRTR database. For the
years 2019 and 2021 DECSO observes high emissions of about 5.5 Gg per year, but this is lower
than the reported value of more than 7 Gg per year. CAMS-TEMPO also shows lower emissions
with a negative trend. Godłowska et al. (2023) showed the stack measurements of this power
plant in their Figure 7, which also are in general lower than the E-PRTR values.
The next selected isolated power plant is the Šoštanj lignite power plant in the Velenje basin
in a mountainous area of Slovenia. It is responsible for one third of the electricity need of
Slovenia (Boznar et al., 2012). For this LPS both CAMS-TEMPO and DECSO show more than
two times higher emissions than E-PRTR, which is too large to be explained by the small cities
or other small sources located in the neighbourhood.
The last case is that of the power plants of the Ptolemais-Amyntheon and Florina coal basins
in West Macedonia, Greece, which were also studied by Skoulidou et al. (2021). There are 5
power plants associated with and located at this basin, but only three are still active: Agios
Dimitrios (1595 MW), Kardia (1200 MW), and Amyntheon (600 MW) (Kostakis, 2009). For
2021 no data was reported for Amynteon in the E-PRTR database. The reported values of the
E-PRTR database match those of CAMS-TEMPO and DECSO quite well, excep for the year 2020
that marks the start of a decrease in emissions in this region. The decreasing trend can be
seen in all three emissions time lines, but is strongest in the E-PRTR time series. Most notable
in the Figure is the strong seasonal cycle in DECSO $NO_x$ emissions for the Greek power plants
with the lowest emissions in summer time. This can related to the availability of more
sustainable energy sources in the summer months.

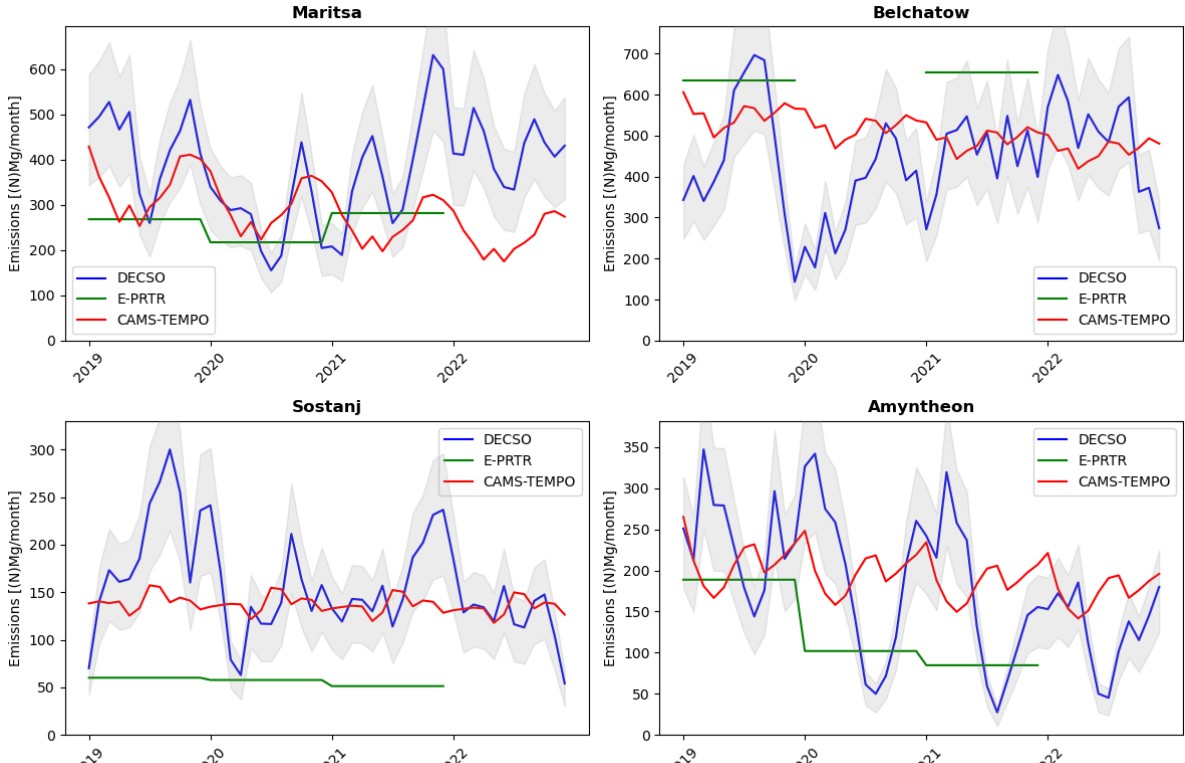


*Figure 7*      *Timeseries of the NO$_x$ emissions of the selected LPS in Europe as estimated by DECSO (blue line), E-PRTR (green line) and CAMS-TEMPO (red line). The shaded grey area shows the estimated uncertainty on the DECSO emissions.*

From this comparison for several large LPS in Europe, we see that CAMS-TEMPO and DECSO are often larger than the reported emissions in E-PRTR. In view of the completely different methodologies and the estimated precision of 25 % for DECSO monthly emissions, the annual values of CAMS-TEMPO and DECSO are often in reasonable agreement (within 20%), but the variability of DECSO is much higher than of CAMS-TEMPO. Emissions of thermal power plants are more intermittent because of the variability of energy demand and variability in energy supply introduced by solar and wind energy sources (Kubik et al., 2012). Note also that CAMS-TEMPO has the exact same seasonal variability for each of the 4 years, which seems unrealistic. The CAMS-TEMPO emissions in the period 2019 to 2022 show for most studied LPS a constant negative trend, which was generally not detected in DECSO. Without additional information it is difficult to draw any conclusions on the performance for LPS, but DECSO supplies additional information on these industrial facilities in Europe and the largest discrepancies may be caused by strong diurnal variability (while TROPOMI observes at about 13:30) and will be interesting for further investigation.

In all cases we see lower emissions in 2020 during the COVID-19 pandemic. In this period the demand of energy was lower and while renewable energy output remained similar, the energy from lignite-based power plants was in relatively less demand (Quitzow et al., 2021).

**Table 1**      Annual NO$_x$ emissions (N)Gg/year of the four lignite power plants. CAMS in the table refers to CAMS-TEMPO.

| Facility | 2019 | | | 2020 | | | 2021 | | | 2022 | | |
|---|---|---|---|---|---|---|---|---|---|---|---|---|
| | CAMS | DECSO | E-PRTR | CAMS | DECSO | E-PRTR | CAMS | DECSO | E-PRTR | CAMS | DECSO | E-PRTR |
| | Unit: (N)Gg/yr | | | Unit: (N)Gg/yr | | | Unit: (N)Gg/yr | | | Unit: (N)Gg/yr | | |
| Maritsa | 4.1 | 5.2±0.4 | 3.2 | 3.6 | 3.3±0.3 | 2.6 | 3.2 | 4.6±0.4 | 3.4 | 2.8 | 5.0±0.4 | - |
| Belchatow | 6.6 | 5.5±0.4 | 7.6 | 6.3 | 4.3±0.3 | - | 5.9 | 5.4±0.4 | 7.9 | 5.6 | 6.0±0.5 | - |
| Sostanj | 1.7 | 2.4±0.2 | 0.69 | 1.7 | 1.7±0.1 | 0.66 | 1.6 | 1.9±0.2 | 0.62 | 1.5 | 1.3±0.1 | - |
| Amyntheon | 2.5 | 2.8±0.2 | 2.3 | 2.4 | 2.3±0.1 | 1.2 | 2.3 | 2.0±0.2 | 1.0 | 1.6 | 1.3±0.1 | - |

## 4. Discussion

We presented the latest version of the DECSO algorithm, version 6.3. Updates has been made for the superobservations, the chemical transport model, the sensitivity matrix and the error parametrization. The new version also includes an error estimate for the monthly $NO_x$ emission data taking into account the autocorrelation in time. The new DECSO version has been applied to the domain of Europe and show more spatial details than before as a result of the higher resolution of TROPOMI observations compared to earlier satellite observations. In the comparison with CAMS-REG over Europe (where emissions are usually well-known) the deviations are small (within 10%) when looking at country scale. For point sources the spread in the differences is much higher, but no systematic effect is yet found. For cities DECSO show higher emissions, while CAMS-REG is higher for rural regions. On a European scale the biggest difference between CAMS-REG and DECSO was found for the region West of Belgrade in Serbia, where the Nicola Tesla power plants are located. While these show up as a strong emission source close to Belgrade in both the DECSO emissions and the E-PRTR database, they are not included or mislocated in the CAMS-REG emissions. This is a prominent example that demonstrates the value of monitoring emissions with satellite observations.

The precision of the derived emissions by DECSO are given for each grid cell in the data files. In general, we can say that the precision of $NO_x$ emissions given per grid cell (0.2x0.2 degree) is about 8% for annual emissions, 25% for monthly emissions and between 10 and 60 % for the daily emissions. When averaging over a larger domain the precision will of course become higher by the square root of the number of grid cells.

The comparison between CAMS-REG and DECSO emissions showed that DECSO is very similar to CAMS-REG for the spatial distribution and the country totals. While compared to the reported emissions in NEC or LRTAP, DECSO is 7 % higher. Validation of the TROPOMI $NO_2$ observations showed that, when using averaging kernels, the bias of the tropospheric column is estimated as -8% on average by comparison with MAX-DOAS observations (Keppens and Lambert, 2023). This bias of -8% should result in lower emissions by DECSO and the deviation between DECSO and other inventories would be higher in reality. Keppens and Lambert (2023) further report that for polluted regions the mean bias of the TROPOMI $NO_2$ observations is stronger, about -29%, while for clean areas the median bias is positive and about +13% (when using averaging kernels). This would be contradictory to our findings over cities, where DECSO shows higher emissions than CAMS-REG. These lower emissions of CAMS-REG in cities as

compared to the rural regions may point to an underestimation of bottom-up traffic
emissions, but uncertainties in both satellite observations and bottom-up emissions are in
general high. Another potential cause of biases in our emissions is the CHIMERE model. More
research is needed for a better understanding of the validation results of TROPOMI
observations, CHIMERE performance, and the comparisons between DECSO and CAMS.
This study shows the potential of DECSO for operational emission monitoring for Europe. The
monitoring of LPS is only possible for isolated sources, thus a future  improvement can be
made by providing the emissions on a higher resolution at the cost of longer processing time.
This will allow the study of more isolated LPS. DECSO has already demonstrated its
performance on a $0.1°\times0.1°$ for smaller regions like the Yangtze River Delta (Zhang et al.,
2023), West Siberia (van der A et al., 2020) and the Netherlands.
In this study the focus was on Europe, but in other regions of the world emissions might be
less well-known. For these regions DECSO can or has been applied since we have global
satellite observations. Recently we have applied DECSO to areas in Africa, where several mines
with high $NO_x$ emissions were found that were unreported in bottom-up emission inventories
like EDGAR or CAMS. This shows the possibilities also for application of DECSO in the Global
South.

**Data availability**
The TROPOMI NO2 data version 2.4 are available via the Copernicus website
https://dataspace.copernicus.eu/ and via the TEMIS website
https://www.temis.nl/airpollution/no2.php (https://doi.org/10.5270/S5P-9bnp8q8).
The $NO_x$ emissions of DECSO v6.3 are available on the GlobEmission website:
https://www.temis.nl/emissions/region_europe/datapage_nox.php.
The European emissions data sets for countries NEC, LRTAP and large facilities E-PRTR are available
on the website https://www.eea.europa.eu/en/analysis/ of the EEA.
The CAMS databases CAMS-REG-ANT v5.1 and CAMS-GLOB-TEMPO v3.1 are available on the ECCAD
website on respectively https://eccad.sedoo.fr/#/metadata/608/ and.
https://eccad.sedoo.fr/#/metadata/504/ (DOI:10.24380/ks45-9147).

**Author contributions**
RA and JD made the improvements to DECSO, HE developed the superobservation code. RA
did the processing, visualisations and main writing. JD and HE reviewed and edited the
manuscript.

**Competing interests**
The authors declare that they have no conflict of interest.

**Acknowledgments**
This research was part of the Sentinel EO-based Emission and Deposition Service (SEEDS,
Grant ID 101004318) project that has received funding from the European Union's Horizon
2020 research and innovation programme. Sentinel-5 Precursor is a European Space Agency
(ESA) mission on behalf of the European Commission. The TROPOMI payload is a joint
development by ESA and the Netherlands Space Office. The Sentinel-5 Precursor ground
segment development has been funded by ESA and with national contributions from the
Netherlands, Germany, and Belgium. This work contains modified Copernicus Sentinel-5P
TROPOMI data (2018–2023), processed locally at KNMI.

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
