# Peer review of "Monitoring European anthropogenic NOx emissions from space"

_EGUsphere, 2023_

## Referee Comment (RC2)

**Review of "Monitoring European anthropogenic NOx emissions from space**

The authors describe the application of the DECSO inversion algorithm to obtain NOx emissions from the TROPOMI instrument over Europe, then compare the results with various emission inventories. The comparisons are done by country, over several megacities and some large point sources. The paper is mostly well organized, though in few places some plot descriptions are rather abruptly inserted. There are few issues that should be resolved before publication. First, while both the DECSO approach and the TROPOMI instrument are well described elsewhere, a bit more information here would be very helpful (see my comments for lines 117 and 195). Moreover, as the method description is so short, one would expect more example and analysis in an ACP paper, but the authors present country totals, three megacity maps and four large point source time series in the main body of the paper. The seems a bit light for ACP and I have suggested additional areas to analyze. Finally, while the authors describe how DECSO error estimates are generated, they are only presented in a very broad sense and I have requested that the DECSO errors be included in tables and plots.

**Major revisions**

Lines 188-189: The authors state that agricultural emissions are excluded in CAMS-REG and in DECSO. For all DECSO emission retrievals? Are they also excluded in NEC, LRTAP and E-PRTR? Please clarify and justify this exclusion.

Line 208: Please explain why the DECSO total emissions agree better with CAMS-REG than with the NEC and LRTAP. Could it be due to the higher spatial resolution provided by CAMS-REG?

Line 226: Authors should expand this section over not just Europe's largest cities but also large industrial areas, such as the Ruhr and Po valleys. And move the Serbian example into this section. Greater London, Greater Amsterdam and Istanbul would also be interesting.

Line 221: Why not add CAMS-REG emissions to this plot?

Line 250: The time series plot (S1) is very interesting. Please provide similar plots for the other cities/regions analyzed and put them in the main body of the paper, not in a supplement.

Line 278: The differences between the various emission sources are not small at all. Does the DECSO uncertainty encompass the CAMS values? See comment on for line 323.

Line 304: In all four cases DECSO shows much more temporal variability than the other two emission estimates. Please present possible sources for this difference in variability. Maybe the temporal resolution? Or is DECSO measuring emissions not included in CAMS? Please comment.

Line 323: This table and the preceding section would greatly benefit from some error analysis. The authors describe how DECSO uncertainty values are generated and present general error estimates in the discussion section, but errors should be included in the table and on the plots.

**Minor changes**

Line 54: it **only** provides

Line 55: biases**,** especially

Line 63: events, for example **(omit like)**

Line 88: please explain what persistency from the analysis means

Line 117-118: Please define and reference TM5-MP model and provide an equation (or equations) that shows how the model and satellite data are combined.

Line 118: in the **satellite L2** file.

Line 142: Please expand a bit on why this assumption is valid.

Line 174: Please provide the temporal resolution of the CAMS and E-PRTR emissions.

Line 195: Please provide a short description of the TROPOMI instrument: launch date, spectral and spatial resolution, swath width and the characteristics of the NO2 product (frequency used, expected error).

Line 281: In what country is the Belchatow power plant located?

---

## Author Comment (AC1)

**Response to review**

We thank the reviewers for their time spent to help us improving the paper with their useful suggestions. Below we provide a point-by-point response to the reviews and descriptions how we have addressed the suggestions. The original review text is indicated in italic blue font and the response in regular black font. New text appearing in the revised paper is indicated by a red color.

**Reviewer 1**

*Van der A et al. discuss in their manuscript a new version of the DESCO NOx emission data set, based on TROPOMI satellite NO2 measurements. The manuscript discusses a valuable data product and certainly deserves to be published after addressing the points mentioned below in a major revision. In particular the structure and the internal logic of the manuscript should be improved. I think this manuscript could be considered within the scope of ACP, but for my taste, it would better fit into AMT, since it is mainly discussing a new data product.*

Since the methodology of DECSO is already addressed in several earlier papers, in this manuscript we focus more on the results for Europe, and therefore we have chosen to submit this manuscript to ACP. In addition, we have added more results and analysed areas on request of reviewers, which makes it even more suitable for ACP.

**General points:**

*- The discussion about uncertainties, in particular for DESCO, but with the comparisons with other data sources also their error estimation, are out of place (see below), and not very extensive. Some numbers are given, but I would like to know, how the authors received these numbers.*

We have addressed extensively the calculation of error estimates for DECSO in section 2.1, which are used in the uncertainty numbers throughout the paper. We have however added more uncertainty estimates in the Table and Figures of the revised manuscript. Error estimates for bottom-up inventories (other data sources) are unfortunately not available. Further clarifications will follow below.

*- The DESCO algorithm uses a Kalman filter, so I think the description of DESCO should mention the influence of the a priori to the results.*

We agree and have added a description of DECSO and the use (or not) of a priori :

Although HTAP v3 has been used for the sector distribution of emissions and other species in CHIMERE, no use is made of a -priori (bottom-up) $NO_x$ emissions in DECSO. DECSO is using a persistency forward model in which the emissions of the current day are equal to the emissions of the previous day. In addition, there is a strong dependency of the calculated emissions on the

observations as shown in Ding et al. (2021). Since the derived emissions are updated by addition and not by multiplication factors, unknown sources or emission changes are detected fast.

*- Cloud filters are mentioned briefly when the TROPOMI measurements are introduced. However, I would be interested in the fraction of filtered measurements, and if any bias due to the fact that one needs to do cloud filtering are expected.*

The fraction of cloudy pixels is about 55% over Europe. However, all available observations within 150 km downwind of the source are used in DECSO, which means that the emissions are almost daily adjusted. For clarification we added (line 154-155):

The observations with a cloud radiance fraction of more than 50% (this corresponds to a cloud fraction of about 20%) have not been used. For Europe, it means that about 45% of the observations are used.

and at line 111-113:

With this sensitivity matrix not only observations over the source are affecting the derived emissions, but also the transported concentrations away from the source within 150 km.

*- In some of the figures, the font size should be increased for better readability.*

We increased the font size of many Figures in the paper.

*- The term NOx is very important in this manuscript and thus appears quite often in the text. Please make sure that it is spelled consistently. I think it should be with a subscript x (also in the title).*

Yes, we have made it more consistent by correcting this everywhere.

*- The section "Discussion" is very strangely structured. Most of the information here appears for the first time, which is not adequate for such a section (in my opinion). The parts about Belgrade, error estimations and the comparison description should be moved to the corresponding sections above - I really was desperately looking for some of these information while reading the manuscript for the first time.*

We agree that the discussion on the region around Belgrade should be moved to the main text. We have both the discussion and Figure S2 about Belgrade moved the main sections.

However, the discussion on the error estimate is a summary of what was already discussed in section 2.1 and has no new information. The discussion on the comparison is a wrap-up of what was earlier discussed.

*- The usage of the term CAMS is not very clear throughout the manuscript. It is mentioned that there are two different CAMS products used (CAMS-REG and CAMS-TEMPO), and I think the manuscript should be precise enough to tell the right product everywhere in the text.*

We had in the original text omitted the specific product (REG or TEMPO) when it was clear from the context which product was meant. In the revised manuscript, we have specified the type for each every time the CAMS product is mentioned.

*- The usage of the "other" data sources than DESCO is not clear to me. Are they considered to be a benchmark, DESCO is compared with? Or are these data known to be uncertain and DESCO should be used to improve such data? Both approaches are used in different parts of the manuscript, even though, these approaches are contradictory.*

We present an approach for deriving NOx emissions that is very different than the bottom-up approach. The methods are very complementary and we can learn a lot by simply comparing these emission results. The strength of DECSO is that it is based on observations, but all the presented methods have similar uncertainties.

Note that large-scale in-situ measurements of emission do not exist and therefore a validation as is used for concentration.

*Specific points:*

*L27: "Knowledge of NOx emissions...": Isn't this sentence redundant to the previous one? If not, please explain why NOx emissions are important for climate studies.*

We have combined the sentence into one sentence to remedy this.

*L29: Regulations of NOx concentrations and emissions are not just present in Europe (e.g. https://www3.epa.gov/region1/airquality/nox.html). I see that the focus of this manuscript is on Europe, but then the sentence should be formulated in a different way.*

We moved "in Europe" to the front of the sentence to emphasize the focus on Europe in this paper. We focus on Europe here, so we think it not relevant to discuss air quality regulations in other parts of the world.

*L44: Suggestion to rephrase: "observation based" -> "based on observations"*

We have corrected this.

*L48: I think, the authors are only referring to satellites in nadir geometry here (which are suited best to derive sources), so this should be mentioned. Further, I stumbled a little about the term "total concentrations"; do the authors mean "total columns" instead?*

Yes, we changed this to "nadir-viewing satellites" and "total column concentrations".

*L59: "For a full three dimensional description...": Out of curiosity: How good is the vertical resolution of typical NO2 (nadir) satellite measurements, and does that vertical resolution allow comparisons to 3d models?*

The satellite observations have little or no information on the vertical profile of $NO_2$. The observations are only used to derive vertical columns of $NO_2$.

*L65: "Interesting aspects ...": Are these aspects discussed in this paper? Otherwise I would skip this sentence, or make it more clear that these are important issues which are beyond the scope of this paper.*

These aspects were mentioned in the referred papers of the previous sentence, so we made this more specific: "Guevara et al. (2021) and Cripa et al. (2023) concluded that interesting aspects for future studies are the spatial distribution, seasonal time profiles and multi-annual trends of the emissions.

*L96: "The most important updates are ...": For my taste this sentence posed a lot of questions, which were answered in the next paragraphs, so I would suggest to either omit this sentence or refer to the following text.*

We added: "More details of these updates follow below."

*L100: Is CHIMERE an acronym? If so, please define. Same for HTAP and ECMWF later on.*

CHIMERE is not an acronym. We have added the acronym definition and a reference for HTAP and the definition of ECMWF.

*L107: Maybe I'm not the expert here, but I don't understand what is meant by "a range of maximum 150 km" in this context. Is it the maximum path the trajectories are allowed to travel? Similar for the temporal resolution "(maximum 7.5 minutes)": Is that the maximum time the trajectories are calculated?*

We have tried to clarify this by changing the text into " The sensitivity matrix, giving the relationship between emissions and concentrations, is based on trajectories calculated with a high temporal resolution (time steps of 7.5 minutes). In the new version the relationship between emissions and concentrations is limited to a maximum distance of 150 km to avoid effects of errors in the trajectories over longer distances. With this sensitivity matrix not only observations over the source are affecting the derived emissions, but also the transported concentrations away from the source within 150 km."

*L113: Suggestion: "to the simultaneously" -> "to simultaneous"*

Corrected.

*L118: Please define TM5-MP*

*L119: I don't understand the sentence "The corresponding averaging kernels..." Please rephrase.*

We replaced the related text by the following explanation:

The modelling of $NO_2$ in the free troposphere, governed by processes like lightning, deep convection, aircraft emissions or long-range transport, is often simplified in regional air-quality models focusing on surface concentrations. However, the TROPOMI $NO_2$ product is providing a tropospheric column, which includes the Planetary Boundary Layer (PBL) and the free troposphere. As a result, model biases in the free troposphere may be a significant source of systematic error in the model-satellite comparisons (Douros et al., 2023). To mitigate this problem we adapt the TROPOMI $NO_2$ retrieval by replacing the tropopause level by a 700 hPa level. The stratosphere + free troposphere $NO_2$ column from the TM5-MP (Tracer Model 5, https://tm5.site.pro/, Williams et al., 2017) assimilation system are now subtracted from the satellite-observed total column, and new retrieved layer column amounts, air-mass factors and kernels are computed for the surface to 700 hPa layer in the same way as they are computed for the tropospheric column (van Geffen et al., 2022b).

*L138: "this new software will be used in future DESCO studies.": I'm confused: Is the newest version of the "software" used in this study or is it only planned to be used in future studies?*

We changed this into " …this new superobservation software is planned to be used in future DECSO studies."

*L170: Figure 1 is only briefly mentioned in the text, but not further discussed at all. I think it very interesting to finally see some results, but the authors could at least discuss a little that the results are plausible (point out bigger cities or ship routes).*

We agree and have added more discussion:

"In the Figure the emissions of major cities and industrial facilities can be identified. Ship emissions show up clearly in most seas where many ships follow the same route. Other areas over sea appear noisier since ship locations are moving while emitting $NO_x$. The most polluted regions in Europe are the densely populated and industrial regions in the Po Valley, the Ruhr area, and the West of the Netherlands."

*L188: "the agricultural emissions, which are also excluded in DECSO.": How does DESCO exclude agricultural emissions? I think it was only mentioned that biogenic and anthropogenic emissions are separated?*

We adapted the text by changing it to : "exclude the soil emissions (i.e. agricultural categories), since soil emissions are also excluded in DECSO". The term "biogenic emissions" are replaced by "soil emission" throughout the document to be more precise. The description to split emissions into soil and anthropogenic emissions has also been extended to explain more about the method (line 175-180).

*L208: I did not understand the reason why CAMS-REG data was not included into Figure 2, while it was mentioned in the text that the total emissions for the examined region matches well with DESCO.*

Yes, it is better to include the CAMS-REG emission too. We have corrected this omission with a new Figure (Fig.2) including CAMS-REG.

*L221: I think that Figure 2 is really hard to read: The interesting thing here are the differences between DESCO and LRTAP and NEC. So the authors could consider to add a (relative or absolute?) difference plot here, where also the uncertainties of each emission could be added. At the moment, I don't know if the agreement is good or not (only Italy pops out, as mentioned in the text). But are the differences of e.g. France and the Czech Republic notable or not?*

A new plot (Figure 2b) has been added made to show the absolute differences of all inventories compared to DECSO.

*L214: "since these emissions are weighted by the land fraction": Why are there then any ship tracks visible in Figure 1? The weighting with land fraction should be mentioned in Section 2, in my opinion.*

This land fraction is only used in the summing of country totals (of Fig.2), since in general country totals are excluding maritime emissions. We added a clarification:

"… since we have weighted the emissions by the land fraction in each grid cells to exclude maritime emissions in these country totals."

*L229: Are Madrid and Rome really megacities? In my understanding this is for cities of more than 10 million inhabitants in the greater area. I would suggest to rather use the term "large city". Further, these three cities are not the largest in Europe (as mentioned in the text) - at least London is within the region of interest and considerably larger in population.*

We changed the term megacities to large cities. We have also added London to the cities tha are discussed.

*L230: It is stated that the country total emissions are similar for DESCO and CAMS-REG (even though this was not shown before), but in all cases absolute emissions are higher for the cities in DESCO. To my understanding this also means, that DESCO needs to be smaller compared to CAMS-REG, in order to have comparable country total emissions. So at what places is DESCO smaller than CAMS-REG for these countries? Are there systematic differences? Is this information interesting to improve the CAMS-REG data?*

The rural regions are in general lower in DECSO, which partly compensated the cities. We have clarified this, by adding the following new text:

The map of the greater area of London shows that DECSO has higher emissions in the city, but lower outside the city. This is a pattern we see in general: in most big cities the emissions derived by DECSO show a similar pattern than in CAMS-REG but the emission are higher, the emission in rural regions on the other hand are usually lower in DECSO than in CAMS-REG. The lower emissions in the rural regions can be seen in Figure S1, which show maps for Europe of both emission products.

*L250 and following: Again, there are huge differences between DESCO and CAMS. Sometimes the trends in the time series seem to be anti-correlated (e.g. August to November 2019: DESCO shows a decreasing trend, while CAMS shows an increase). This should be discussed. Further, I find it really hard to identify the mentioned "3 low emission periods". Maybe these periods could be marked in Figure S1? Additionally, it was said that the biogenic, time dependent signal was removed from the DESCO data. So in Figure S1, only the anthropogenic data is shown, but still has a time dependency?*

We have moved Figure S1 to the main text on request of reviewer 2 (it becomes Figure 4) and we have added timeseries for 3 other cities and more text to discusses the timeseries:

Figure 4 shows examples of timeseries for city emissions, in this case for the cities of Paris, Madrid, Istanbul and Rome (also shown in Figure 3a). In these plots we report the total emissions in a square area of 5 by 5 grid cells centred on the city centre to make sure the whole city has been captured. As we had seen earlier, the DECSO emissions are on average higher than for CAMS-TEMPO, but also the seasonal cycle is different. The $NO_x$ emissions of CAMS-TEMPO show a seasonal cycle, which is almost identical each year, while DECSO show larger variations from year-to-year. We see clearly the effect of COVID regulations in all cities, that started first in March/April 2020 in Europe, and in the winter of 2020-2021 when strict COVID regulations were again in place. The general overall trend in this 4 year time period varies from city to city, but most cities show a slightly decreasing trend, partly related to a gradual decrease of emissions from road vehicles linked to European regulations.

*L276: "The difference in annual emissions [...] are relatively small ...": I do not agree with this statement at all. For 2019, DESCO is 1.1 Gg(?) higher than CAMS, and 2 Gg(?) higher than E-PRTR. This is 20%-40% relative difference, which I would never call "relatively small"!*

We changed this "relatively small" to "within 20-40%".

*L323: Table 1 gives only units in the caption, I would prefer to see them also at least in each column heading. However, the unit (N)Gg/year is a bit strange, since yearly values are shown here. Should this not rather be (N) Gg, since it is already multiplied by the time of one year?*

The unit has been added in Table 1. However, the unit for emissions should always include the time dimension. This is analogue to the average speed in an hour time interval given in km/h (or m/s).

*Section 3.3: For most of the discussed LPS, there is little discussion about the comparison between DESCO and the other data sources, apart from the Maritsa Iztok facility. I would like to learn more about the (paritally quite substantial) differences, and why there is more month-to-month variability in DESCO compared to CAMS-TEMPO.*

Unfortunately, there is little information to be found about the production or emissions from these facilities besides the numbers in the E-PRTR database. We want to show that although bottom-up inventories lack often information on the seasonal cycle, DECSO can provide this information independently, while the annual emissions are not very different. We have changed the text:

In view of the completely different methodologies and the estimated precision of 25 % for DECSO monthly emissions, the annual values of CAMS-TEMPO and DECSO are often in reasonable agreement (within 20%), but the variability of DECSO is much higher than of CAMS-TEMPO. Emissions of thermal power plants are more intermittent because of the variability of energy demand and variability in energy supply introduced by solar and wind energy sources (Kubik et al.,

2012). Note also that CAMS-TEMPO has the exact same seasonal variability for each of the 4 years, which seems unrealistic.

*L312: Why are differences of 20% a "reasonable agreement" in this study? As mentioned above: A discussion of error estimates would be very important here.*

Since the methodologies are very different: CAMS is based on reported information from the facilities and emission factors, while DECSO is based on measurements and have a precision of 25 %. In that situation we would call 20% deviation reasonable. We have adapted the text and mentioned the precision.

*L336: "... while CAMS is higher for rural regions.": I think I missed this information in the main part of the manuscript?*

We have added this in the discussion of Figure 3 and 2 (eg. Line 204-305). In addition we showed a new Figure (Figure S1) in the supplement to show the differences between CAMS-REG and DECSO for the whole of Europe.

*L347: "very similar on average." I would disagree on that statement given the larger descrepancies for the cities and LSPs. Maybe that is meant to be for the country emissions, but that should be stated.*

We rephrased this to: "The comparison between CAMS and DECSO emissions showed that DECSO is very similar to CAMS for the spatial distribution and the country totals"

*L353 Keppens and Lambert -> Keppens and Lambert (2023)*

Corrected.

*L361: "only gives mainly clear results": This is an awkward formulation, please rephrase.*

We have rephrased this to: "The monitoring of LPS is only possible for isolated sources,.."

*L362: "improvement can be gained by providing the emissions on a higher resolution.": So why has this not been done in this study? I'm a little disappointed to learn in the end of the manuscript, that there have been studies with better resolution than the one I just spent some time with.*

We rephrased this sentence to make clear that the high resolution has only been applied for smaller regions since the processing is expensive. The adapted text is as follows:

This study shows the potential of DECSO for operational emission monitoring for Europe. The monitoring of LPS is only possible for isolated sources, thus a future improvement can be made by providing the emissions on a higher resolution at the cost of longer processing time. This will allow the study of more isolated LPS. DECSO has already demonstrated its performance on a 0.1°x0.1° for smaller regions like the Yangtze River Delta (Zhang et al., 2023), West Siberia (van der A et al., 2020) and the Netherlands.

*L372 and following: The URLs to the websites should all be included as clickable links within the PDF. Some of them are clickable, but broken (but the plain text works). It would be even better if the data sets would have DOIs, but that is most likely out of control for the authors.*

We have corrected the links and added DOIs when available.

**Reviewer 2**

*The authors describe the application of the DECSO inversion algorithm to obtain NOx emissions from the TROPOMI instrument over Europe, then compare the results with various emission inventories. The comparisons are done by country, over several megacities and some large point sources. The paper is mostly well organized, though in few places some plot descriptions are rather abruptly inserted. There are few issues that should be resolved before publication. First, while both the DECSO approach and the TROPOMI instrument are well described elsewhere, a bit more information here would be very helpful (see my comments for lines 117 and 195). Moreover, as the method description is so short, one would expect more example and analysis in an ACP paper, but the authors present country totals, three megacity maps and four large point source time series in the main body of the paper. The seems a bit light for ACP and I have suggested additional areas to analyze. Finally, while the authors describe how DECSO error estimates are generated, they are only presented in a very broad sense and I have requested that the DECSO errors be included in tables and plots.*

Thanks for your suggestions. At first we had kept the method description short, since it was already described in previous papers. However, we have now added more details about the methodology and the TROPOMI instrument. In addition, more examples and analyses are given and DECSO errors are included at more places.

**Major revisions**

*Lines 188-189: The authors state that agricultural emissions are excluded in CAMS-REG and in DECSO. For all DECSO emission retrievals? Are they also excluded in NEC, LRTAP and E-PRTR?*

*Please clarify and justify this exclusion.*

We have split the DECSO emission into anthropogenic emissions and soil emissions and in this paper we discuss only the anthropogenic emissions. For a fair comparison we also excluded these emissions in the other inventories (NEC, LRTAP, E-PRTR, CAMS-REG, CAMS-TEMPO). We have clarified this at several places in the document by using more consistent terminologies.

*Line 208: Please explain why the DECSO total emissions agree better with CAMS-REG than with the NEC and LRTAP. Could it be due to the higher spatial resolution provided by CAMS-REG?*

The difference between CAMS-REG and NEC/LRTAP is due to fact that the methods are very different. NEC/LRTAP are based on the officially report emissions, which is also input for CAMS-REG, but more processing is done in CAMS-REG to fil in gaps or correct data. The spatial resolution of CAMS-REG should not affect the country totals.

Since DECSO is in all aspects a completely different method than the bottom-up methods, we cannot explain why DECSO better agrees with CAMS-REG.

*Line 226: Authors should expand this section over not just Europe's largest cities but also large industrial areas, such as the Ruhr and Po valleys. And move the Serbian example into this section. Greater London, Greater Amsterdam and Istanbul would also be interesting.*

We have followed this suggestion by adding new Figures for these regions. Figures for Greater London, Istanbul, Po Valley, Ruhr, Serbia have been added in Figure 3a, 3b and 3c.

*Line 221: Why not add CAMS-REG emissions to this plot?*

We agree and we have added the CAMS-REG emissions in Figure 2. This was also suggested by reviewer 1.

*Line 250: The time series plot (S1) is very interesting. Please provide similar plots for the other cities/regions analyzed and put them in the main body of the paper, not in a supplement.*

We have added in a new Figure 4 (the old Fig4 is now Fig5) these time series for four cities that were already shown in Figure 3 for the spatial distribution. The text discussing this Figure has been adapted to:

Figure 4 shows examples of timeseries for city emissions, in this case for the cities of Paris, Madrid, Istanbul and Rome (also shown in Figure 3a). In these plots we report the total emissions in a square area of 5 by 5 grid cells centred on the city centre to make sure the whole city has been captured. As we had seen earlier, the DECSO emissions are on average higher than for CAMS-TEMPO, but also the seasonal cycle is different. The $NO_x$ emissions of CAMS-TEMPO show a seasonal cycle, which is almost identical each year, while DECSO show larger variations from year-to-year. We see clearly the effect of COVID regulations in all cities, that started first in March/April 2020 in Europe, and in the winter of 2020-2021 when strict COVID regulations were again in place. The general overall trend in this 4 year time period varies from city to city, but most cities show a slightly decreasing trend, partly related to a gradual decrease of emissions from road vehicles linked to European regulations.

*Line 278: The differences between the various emission sources are not small at all. Does the DECSO uncertainty encompass the CAMS values? See comment on for line 323.*

We have rephrased this comment and added more discussion about the error estimates:

From this comparison for several large LPS in Europe, we see that CAMS-TEMPO and DECSO are often larger than the reported emissions in E-PRTR. In view of the completely different methodologies and the estimated precision of 25 % for DECSO monthly emissions, the annual values of CAMS-TEMPO and DECSO are often in reasonable agreement (within 20%), but the variability of DECSO is much higher than of CAMS-TEMPO. Emissions of thermal power plants are more intermittent because of the variability of energy demand and variability in energy supply introduced

*Line 304: In all four cases DECSO shows much more temporal variability than the other two emission estimates. Please present possible sources for this difference in variability. Maybe the temporal resolution? Or is DECSO measuring emissions not included in CAMS? Please comment.*

E-PRTR has only annual emissions, while the CAMS seasonal cycle is derived per country per sector. To explain the variability of DECSO emissions of power plants, we have added:

Emissions of thermal power plants are more intermittent because of the variability of energy demand and variability in energy supply introduced by solar and wind energy sources. (Kubik et al., 2012)

*Line 323: This table and the preceding section would greatly benefit from some error analysis. The authors describe how DECSO uncertainty values are generated and present general error estimates in the discussion section, but errors should be included in the table and on the plots.*

We agree and have added the estimated error of DECSO to the numbers in both the Table and in the plot (Figure 5).

**Minor changes**

*Line 54: it **only** provides*

*Line 55: biases, especially*

*Line 63: events, for example **(omit like)***

This has been corrected in the new manuscript.

*Line 88: please explain what persistency from the analysis means*

We have added a short explanation: "…using a persistency forward model in which the emissions of the current day are equal to the emissions of the previous day"

*Line 117-118: Please define and reference TM5-MP model and provide an equation (or equations) that shows how the model and satellite data are combined.*

We replaced the text with the following explanation including reference for TM5-MP:

The modelling of $NO_2$ in the free troposphere, governed by processes like lightning, deep convection, aircraft emissions or long-range transport, is often simplified in regional air-quality models focusing on surface concentrations. However, the TROPOMI $NO_2$ product is providing a tropospheric column, which includes the Planetary Boundary Layer (PBL) and the free troposphere. As a result, model biases in the free troposphere may be a significant source of systematic error in the model-satellite comparisons (Douros et al., 2023). To mitigate this problem we adapt the TROPOMI $NO_2$ retrieval by replacing the tropopause level by a 700 hPa level. The stratosphere + free troposphere $NO_2$ column from the TM5-MP (Tracer Model 5, https://tm5.site.pro/, Williams et al., 2017) assimilation system are now subtracted from the satellite-observed total column, and new retrieved layer column amounts, air-mass factors and kernels are computed for the surface to 700 hPa layer in the same way as they are computed for the tropospheric column (van Geffen et al., 2022b).

*Line 118: in the **satellite L2** file.*

Corrected.

*Line 142: Please expand a bit on why this assumption is valid.*

It is actually not an assumption, but rather an observation, so we changed the text. We added a short explanation of the method before we refer to the paper under review that is describing the full method:

The soil $NO_x$ emissions are derived by fitting the monthly emissions in a selection of grid-cells without any significant anthropogenic contribution according to land-use data. In this way the monthly averaged soil $NO_x$ emissions in the categories for forest, agricultural and shrub-land are derived. These monthly soil $NO_x$ emissions are weighted with the land-use type of these 3 categories in each grid cell and subtracted from the total derived $NO_x$ emissions to end up with the anthropogenic $NO_x$ emissions discussed in this study.

*Line 174: Please provide the temporal resolution of the CAMS and E-PRTR emissions.*

We have added the temporal resolution which is annual, except for the monthly CAMS-TEMPO emissions.

*Line 195: Please provide a short description of the TROPOMI instrument: launch date, spectral and spatial resolution, swath width and the characteristics of the NO2 product (frequency used, expected error).*

We have added:

TROPOMI is a spectrometer instrument onboard the Sentinel 5P satellite, which was launched in October 2017 and is flying a sun-synchronous polar orbit with a local overpass time of 13:30. The measured $NO_2$ columns are derived from the visible band that has a spectral resolution of 0.54 nm (0.2nm sampling) and a signal-to-noise ratio of about 1500 (van Geffen et al., 2022a). The $NO_2$ tropospheric columns have a spatial resolution of 5.5 x 7 km (5.5 x 3.5 km since 6 August 2019) over a swath of about 2600 km, which means that global coverage is reached daily.

*Line 281: In what country is the Belchatow power plant located?*

We have added Poland.

---

## Author Response (AR2)

**Author's reponse**

We have made small changes to the manuscript based on the various suggestions:

**Reviewer 1 (Report #2)**

We have implemented all the technical corrections suggested by reviewer 1.

**Reviewer 2 (report 1#)**

No further corrections were requested.

**File validation**

Instead of putting the figures Figure 3a, 3b, and 3c together into a single big figure, we have split them in separate figures: Figure 3, Figure 4 and Figure 5, and updated all references to these and the following Figures.

The cited references that were in review have been replaced by their preprints including DOI.

**Editor report**

Thanks for these suggestions. The mistake in Figure 5 has been corrected.

Concerning the comparison between DECSO and CAMS, it is actually only the city emissions that are clearly higher in DECSO than in the CAMS-REG and CAMS-TEMPO emissions (Figure 4) considering the uncertainties. We have added the following remark:

"These lower emissions of CAMS-REG in cities as compared to the rural regions may point to an underestimation of bottom-up traffic emissions, but uncertainties in both satellite observations and bottom-up emissions are in general high."